# Activation of Mitochondrial 2-Oxoglutarate Dehydrogenase by Cocarboxylase in Human Lung Adenocarcinoma Cells A549 Is p53/p21-Dependent and Impairs Cellular Redox State, Mimicking the Cisplatin Action

**DOI:** 10.3390/ijms21113759

**Published:** 2020-05-26

**Authors:** Victoria I. Bunik, Vasily A. Aleshin, Xiaoshan Zhou, Vyacheslav Yu. Tabakov, Anna Karlsson

**Affiliations:** 1A.N. Belozersky Institute of Physicochemical Biology, Lomonosov Moscow State University, 119991 Moscow, Russia; aleshin_vasily@mail.ru; 2Faculty of Bioengineering and Bioinformatics, Lomonosov Moscow State University, 119991 Moscow, Russia; 3Department of Biochemistry, Sechenov University, 119991 Moscow, Russia; 4Division of Clinical Microbiology, Department of Laboratory Medicine, Karolinska Institute, Karolinska University Hospital, 141 86 Stockholm, Sweden; Xiaoshan.Zhou@ki.se; 5Common Use Center “Biobank”, Research Center for Medical Genetics, Moscow, Russia; vyutab@yandex.ru

**Keywords:** anticancer effect of cocarboxylase, cisplatin, glutathione, 2-oxoglutarate dehydrogenase, p53, p21, thiamine

## Abstract

Genetic up-regulation of mitochondrial 2-oxoglutarate dehydrogenase is known to increase reactive oxygen species, being detrimental for cancer cells. Thiamine diphosphate (ThDP, cocarboxylase) is an essential activator of the enzyme and inhibits p53–DNA binding in cancer cells. We hypothesize that the pleiotropic regulator ThDP may be of importance for anticancer therapies. The hypothesis is tested in the present work on lung adenocarcinoma cells A549 possessing the p53–p21 pathway as fully functional or perturbed by p21 knockdown. Molecular mechanisms of ThDP action on cellular viability and their interplay with the cisplatin and p53–p21 pathways are characterized. Despite the well-known antioxidant properties of thiamine, A549 cells exhibit decreases in their reducing power and glutathione level after incubation with 5 mM ThDP, not observed in non-cancer epithelial cells Vero. Moreover, thiamine deficiency elevates glutathione in A549 cells. Viability of the thiamine deficient A549 cells is increased at a low (0.05 mM) ThDP. However, the increase is attenuated by 5 mM ThDP, p21 knockdown, specific inhibitor of the 2-oxoglutarate dehydrogenase complex (OGDHC), or cisplatin. Cellular levels of the catalytically competent ThDP·OGDHC holoenzyme are dysregulated by p21 knockdown and correlate negatively with the A549 viability. The inverse relationship between cellular glutathione and holo-OGDHC is corroborated by their comparison in the A549 and Vero cells. The similarity, non-additivity, and p21 dependence of the dual actions of ThDP and cisplatin on A549 cells manifest a common OGDHC-mediated mechanism of the viability decrease. High ThDP saturation of OGDHC compromises the redox state of A549 cells under the control of p53–p21 axes.

## 1. Introduction

Mitochondrial multienzyme complexes of dehydrogenases of 2-oxo acids catalyze irreversible reactions in highly regulated metabolic checkpoints [1,2,3], demonstrating cell-specific expression and functional significance in cancer [4,5,6,7]. In particular, the mitochondrial reaction of oxidative decarboxylation of 2-oxoglutarate is required for glutaminolysis, which is elevated in malignant cells. The reaction also provides succinyl-CoA for the mitochondrial production of the high energy phosphates ATP or GTP at the substrate level, i.e., beyond oxidative phosphorylation [1,2]. Nevertheless, an isoenzyme of 2-oxoglutarate dehydrogenase, encoded by the *OGDHL* gene, is down-regulated in a number of cancers, including the lung adenocarcinoma A549 cells [5], by promoter hypermethylation [8,9,10]. Re-establishment of the *OGDHL* expression in the cancer types with downregulated *OGDHL* gene has anti-proliferative properties associated with the increased production of reactive oxygen species (ROS) by such cells [11]. Unlike the *OGDHL* protein, the ubiquitous *OGDH*-encoded 2-oxoglutarate dehydrogenase is important for A549 cells viability [6,7]. However, this isoenzyme is also known to actively produce mitochondrial ROS under metabolic dysbalance [3]. In view of these data, we hypothesize that pharmacological over-activation of the 2-oxoglutarate dehydrogenase reaction, catalysed by the multienzyme 2-oxoglutarate dehydrogenase complex (OGDHC), may be detrimental for the cancer cells with functional OGDHC. An essential activator of 2-oxoglutarate dehydrogenase is its coenzyme thiamine diphosphate (ThDP), or cocarboxylase. This natural derivative of thiamine (vitamin B1) is a pleiotropic regulator, activating the ThDP-dependent enzymes and inhibiting the DNA binding of the master metabolic regulator p53 [12,13]. Therefore, the goal of this work was to study potential antiproliferative action of the ThDP-dependent activation of the 2-oxoglutarate dehydrogenase reaction in cancer cells and its interaction with the p53 pathway. 

Regarding the goal, it should be noted that ThDP and its precursor thiamine are known to be highly important for metabolically transformed cancer cells [14]. Epidemiological studies suggest that thiamine status is linked to varying cancer rates [15]. Involvement of the ThDP-dependent transketolase into the pentose phosphate pathway, whose oxidative part produces NADPH and is up-regulated in different cancer cell lines under control of p53 [16], is generally supposed to account for an increased thiamine demand by tumors [14]. Yet the relationship between thiamine and cancer is complex and affected by a number of factors. For instance, the breast cancer prognosis in animal experiments depends on the interaction of thiamine supplementation with obesity [17]. Although the involvement of thiamine in central glucose metabolism is important for malignant cells, both up- and down-regulation of the expression of cellular thiamine and ThDP transporters occur in different types of cancers [18,19,20,21,22,23,24,25,26,27], and up-regulation of thiamine diphosphokinase takes place only in cancer cells under hypoxic conditions [28]. Together with the aforementioned down-regulation of the *OGDHL* gene [5,8,9,10,11], available data suggest that a fine tuning of the thiamine-dependent processes in cancer cells is linked to their specific metabolic types. The complexity of the thiamine interaction with cancer metabolism is in line with sporadic observations of the dual action of thiamine on tumor proliferation, which may be not only stimulated, but also inhibited with the low and high doses of thiamine, correspondingly [29,30,31,32]. Despite their therapeutic potential for cell-specific combinatorial therapies, these findings have generally been left unattended, calling upon a more profound study of molecular mechanisms underlying such effects. 

Using the OGDHC activity as an indicator of intracellular ThDP levels, we show that viability of the A549 cells with the fully functional or partly disabled p53–p21 pathway exhibits different response to the ThDP exposure. Our data reveal that ThDP may increase or decrease the viability of A549 cells in a p21-dependent manner, with the p53–p21 axes controlling the OGDHC response to cellular ThDP. In contrast, the viability of a normal epithelial cell line Vero is not reduced in the same concentration interval of ThDP, in good accord with the well-known antioxidant effects of the thiamine supplementation to non-cancer cells and tissues [33,34,35,36,37]. In the current work, we show that ThDP effects on viability of A549 cells depend on the functions of cellular OGDHC and p53–p21 pathway. Moreover, we reveal interaction between the viability-deteriorating actions of ThDP and cisplatin. This finding agrees with the known involvement of cisplatin with p21 [38,39,40] and OGDHC [41], two proteins participating in the ThDP effects on A549 cells too. As a result, we observed similar non-additive effects of ThDP and cisplatin on the viability of A549 cells, pointing to a common OGDHC-mediated mechanism of their actions. The medical relevance of the present work is underlined by our finding that, under conditions of thiamine deficiency, cisplatin, like ThDP, increases the viability of A549^wt^ cells, with the effect abolished by the p21 knockdown. The thiamine deficiency-induced reversal of the cisplatin effect on the viability of the lung adenocarcinoma A549^wt^ cells points to thiamine deficiency as a factor supporting cellular resistance to cisplatin.

## 2. Results

### 2.1. Incubation of A549 Cells with 5 mM ThDP Saturates the Mitochondrial 2-Oxoglutarate Dehydrogenase with ThDP in a p21-Dependent Manner

Endogenous saturation of extractable activity of ThDP-dependent enzymes is known to be an indicator of intracellular ThDP levels [34,42]. Animal OGDHC binds ThDP tightly, not losing the coenzyme upon purification [43]. Therefore, the concentration of the OGDHC-ThDP complex, i.e., OGDHC holoenzyme, in the assay medium without addition of ThDP characterizes the endogenous holoenzyme level inside cells. With 1 mM ThDP added to the OGDHC assay medium, the activity of all available OGDHC (total OGDHC) is measured. The cellular level of the OGDHC apoenzyme, i.e., the enzyme without ThDP bound, is calculated from the difference between the total and holoenzyme activities. 

As shown in Figure 1A, under standard growth conditions, the incubation of A549^wt^ cells with a high (5 mM) concentration of ThDP for 24 h does not influence the total OGDHC activity, pointing to unchanged OGDHC expression. However, there is a significant increase in activity of the OGDHC holoenzyme (Figure 1E), accompanied by disappearance of the OGDHC apoenzyme (Figure 1I). Thus, the incubation of A549^wt^ cells with 5 mM ThDP increases intracellular ThDP, manifest in the complete saturation of mitochondrial OGDHC with ThDP.

Similar to A549^wt^ cells (Figure 1A), A549^p21-^ cells do not exhibit any changes in the total OGDHC activity after 24 h of incubation with 5 mM ThDP added to the standard growth medium (Figure 1B). However, in contrast to A549^wt^ cells, there is no significant reduction in the apo-OGDHC or increase in the holo-OGDHC levels in the A549^p21-^ cells after the incubation with ThDP (Figure 1F,J). Thus, based on the OGDHC saturation with ThDP, A549^p21-^ cells grown in the standard medium respond to the incubation with 5 mM ThDP significantly less than the A549^wt^ cells do.

As a result, in A549 cells, the activation of mitochondrial ThDP-dependent OGDHC by the coenzyme is under control of the p53–p21 axes.

### 2.2. Knockdown of p21 Deregulates Partition of Total OGDHC into the Apo- and Holoenzyme

Upon thiamine deficiency, both A549^wt^ and A549^p21-^ cells exhibit a strong reduction in total activity of OGDHC, compared to that of the cells grown in the standard medium with thiamine (Figure 1C,D vs Figure 1A,B; Table 1). This reduction is accompanied by full transformation of the enzyme to its holoenzyme complex with ThDP, with the OGDHC apoenzyme level approaching zero both in A549^wt^ and A549^p21-^ cells (Figure 1G,H,K,L). When the thiamine deficient A549^wt^ and A549^p21-^ cells are compared, the response to 5 mM ThDP of the total OGDHC, or OGDHC partition between the holo- and apoenzymes, do not significantly depend on p21 (Figure 1). That is to say, the addition of 5 mM ThDP to the thiamine-depleted A549 cells causes a six-fold increase in total OGDHC (Figure 1C,D), accompanied by significant increases in both the holo-OGDHC (Figure 1G,H) and apo-OGDHC (Figure 1K,L) and similar (30–40%) partitioning of total OGDHC into its apo- and holo-forms in both cell lines. This finding is in contrast to the p21-dependent ThDP response of OGDHC in the cells grown under standard conditions (Figure 1, Table 1). Moreover, the type of the response of cellular OGDHC to ThDP in the standard and thiamine deficient media is different. As shown above, in the standard medium the total OGDHC expression does not change in response to ThDP (Figure 1I,J). In contrast, supplementation of the thiamine deficient cells with 5 mM ThDP increases total OGDHC 2-fold (*p* < 0.05 for the black bars in Figure 1A vs Figure 1C and *p* = 0.1 for the black bars in Figure 1B vs Figure 1D). However, a significant part of this ThDP-induced OGDHC, i.e., approximately 40% in A549^wt^ and 30% in A549^p21-^ cells, remains in the form of apoenzyme independent of p21 expression (Figure 1K,L). As a result, upon ThDP addition, A549^wt^ cells in the thiamine deficient and standard media attain similar levels of activity of holo-OGDHC (*p* = 0.7 for the black bars in Figure 1E vs Figure 1G) at significantly different levels of the latent (activatable by ThDP) apo-OGDHC (*p* = 0.004 for the black bars in Figure 1I vs Figure 1K). In contrast, when ThDP is added to A549^p21--^ cells grown in the thiamine deficient or standard media, the cells become strongly different in the levels of active form, i.e., holo-OGDHC (*p* = 0.007 for the black bars in Figure 1F vs Figure 1H), at similar levels of the latent form, i.e., apo-OGDHC (black bars in Figure 1L vs Figure 1J). Hence, A549^p21-^ cells cannot decouple the expression of functional OGDHC from the enzyme saturation by ThDP: Upon 1.4-fold increase in the total OGDHC activity, the OGDHC saturation with ThDP in A549^p21--^ cells increases 1.3-fold (from 55% to 72%) (Figure 1, Table 1). In contrast, A549^wt^ cells are able to decouple the OGDHC expression from the enzyme saturation by ThDP: Upon a two-fold increase in the total activity of OGDHC, the OGDHC holoenzyme portion in A549^wt^ cells not only does not increase, but decreases 1.6-fold (from 96% to 61%) (Figure 1, Table 1).

Thus, under conditions when both the total OGDHC and ThDP levels are high, knockdown of p21 deregulates cellular homeostatic mechanisms which limit the OGDHC activity through the enzyme saturation with ThDP (Figure 1, Table 1).

### 2.3. ThDP Influences Cellular Redox Indicators of A549 Cells in a p5-3 and p21-Dependent Manner

When A549 cells grown in standard medium are incubated for 24 h with 5 mM ThDP, cellular levels of free thiols, mainly represented by glutathione [44], tend to decrease (Figure 2A,B). A comparison of the ThDP-induced changes in cellular levels of glutathione, its disulfide, and their ratio indicates that, in A549^wt^ cells, it is the GSH/GSSG ratio which is decreased by ThDP most significantly (Figure 2I), whereas in A549^p21-^ cells it is GSH itself (Figure 2B). Thus, ThDP added to either A549^wt^ or A549^p21-^ cells perturbs the glutathione homeostasis, but the prevailing mechanisms of the perturbation, such as increased oxidation or decreased biosynthesis of glutathione, depend on p21 expression.

Under thiamine deficiency, A549 cells increase their glutathione level by 50%, as shown by gray bars in Figure 2A vs Figure 2C and Figure 2B vs Figure 2D (*p* < 0.05, Table 1). Simultaneously, glutathione disulfide increases by 50% (*p* < 0.05, Table 1) so that no significant changes in the GSH/GSSG ratio is observed (Table 1). The thiamine-deficiency-induced increase in the glutathione pool is attenuated after 24 hours incubation of the thiamine deficient cells with 5 mM ThDP (black bars in Figure 2A vs Figure 2C and Figure 2B vs Figure 2D). Similar to the cells grown under standard conditions (Figure 2A,B), ThDP addition to the thiamine-depleted cells decreases glutathione in A549^p21-^ cells more significantly than in A549^wt^ cells (Figure 2C,D), whereas the glutathione redox potential is decreased by ThDP in A549^wt^ cells more than in A549^p21-^ cells (Figure 2K,L). Remarkably, compared to standard medium, thiamine deficiency increases the significance of the ThDP effect on the glutathione level in A549^p21-^ cells (Figure 2B vs Figure 2D), decreasing that of the ThDP effect on the glutathione redox potential in A549^wt^ cells (Figure 2I vs Figure 2K). The changes correspond to the ThDP effect on the OGDHC partition into the holoenzyme (Figure 2, Table 1). The higher the %OGDHC-holoenzyme, the more significant a decrease in GSH/GSSG (in A549^wt^ cells) or GSH (in A549^p21-^ cells). Thus, the ThDP-induced perturbations in cellular glutathione pool depend on the OGDHC activity.

In view of the known impact of p53–p21 axes on the glutathione homeostasis [45], we checked if the different mechanisms of the perturbed glutathione homeostasis in A549^wt^ (decreased glutathione redox potential) and A549^p21-^ (decreased glutathione content) cells are linked to the different changes in p53 expression in response to ThDP. As shown in Figure 2M–P, both in the normal and thiamine deficient media, p53 expression in response to ThDP is not changed in A549^wt^ cells, but undergoes a statistically significant increase in A549^p21-^ cells. In contrast, the cell cultivation in the thiamine deficient medium induces app. 2-fold induction of p53 in A549^wt^ cells only (Figure 2O vs Figure 2M, Table 1). Thus, perturbed relationship between the p21 and p53 expression in A549^p21-^ vs A549^wt^ cells causes their different responses to varied thiamine/ThDP levels at the level of p53 induction, linked to the different mechanisms of perturbed glutathione homeostasis (Figure 2) and dysregulated %OGDHC holoenzyme levels (Figure 1).

When cellular reducing potential, estimated by NAD(P)H-dependent reduction of artificial dye XTT (2,3-bis-(2-methoxy-4-nitro-5-sulfophenyl)-2*H*-tetrazolium-5-carboxanilide), is assessed, the ThDP-induced decrease in the redox state is even more pronounced (Figure 2Q,R) than the decrease in glutathione (Figure 2A,B). In fact, both A549^wt^ and A549^p21-^ cells incubated in the medium with 5 mM ThDP show a three-fold decrease in their NAD(P)H:XTT reductase activity (Figure 2Q,R). Thus, a mild (up to 30%) decrease in cellular thiols by 5 mM ThDP is accompanied by a strong (up to 60%) decrease in cellular redox potential (Figure 2A,B vs Figure 2Q,R).

Figure 3 shows that cellular NAD(P)H:XTT reductase activity exhibits a moderate [46] negative correlation with the activity of the OGDHC holoenzyme (Figure 3A), and a moderate positive correlation with the level of the OGDHC apoenzyme (Figure 3B). The negative correlation becomes strong and statistically significant when cellular NAD(P)H:XTT reductase activity is correlated with percentage of holo-OGDHC (Figure 3C).

Although the NAD(P)H:XTT reductase activity is affected more profoundly than the glutathione levels over the multitude of the studied conditions, there is a coupled response of the two parameters to ThDP addition in most cases (Figure 2). The only exception is observed in the thiamine deficient A549^wt^ cells, where cellular NAD(P)H:XTT reductase activity is partly restored by 5 mM ThDP (Figure 2S), while glutathione is not (Figure 2C). This discrepancy correlates with a significant effect of ThDP supplementation to the thiamine deficient A549^wt^ cells on cellular protein (Table 1). In these cells, a 20% decrease of cellular protein is observed, pointing to decreased cell density due to the ThDP treatment, which is less pronounced under other conditions. Usually, increased treatment times (96 hours) or higher concentrations of a drug are required to observe the decreased cell number after the drop in cellular reducing power [5]. Levels of cellular protein in cell lysates indicate that, in the A549^wt^ cells with the normal control of the cell cycle by p21, the decrease in reducing power already starts to be translated into the impaired cell growth after 24 h (Table 1).

While glutathione is calculated per mg of protein, the changes in the NAD(P)H:XTT reductase activity of the cells are relative measures of both the number of cells and their reducing potential. Although the changes in cellular protein after 24 h of the ThDP treatment reach statistical significance only in the thiamine deficient A549^wt^ cells, a less significant 15% reduction in cellular protein also occurs in the ThDP-treated A549^wt^ cells in standard DMEM (Table 1). As a result, under most of the employed conditions, decreases in cellular NAD(P)H:XTT reductase activity after 24 h of treatment with ThDP (Figure 2Q,R,T) are primarily due to the drop in cellular reducing power, manifesting in the levels of cellular glutathione as well.

The concentration dependence of the ThDP effect on the NAD(P)H:XTT reductase activity (Figure 4) also reveals a biphasic response of the thiamine deficient A549 cells to increasing ThDP concentrations. The biphasic kinetics is most pronounced in the thiamine deficient A549^wt^ cells (Figure 4C, the light-grey bar), where a low (0.05 mM) concentration of ThDP restores cellular NAD(P)H:XTT reductase activity to the control level, followed by a partial decrease of this restoration at a higher (5 mM) ThDP (Figure 4C, the dark-grey bar). In the thiamine deficient A549^p21-^ cells, the restoration of the NAD(P)H:XTT reductase activity by 0.05 mM ThDP is not pronounced (Figure 4D, the light-grey bar), and at a higher (5 mM) ThDP concentration, a decrease in the activity is observed compared to the non-treated thiamine deficient cells. As noted above, the biphasic changes in the NAD(P)H:XTT reductase activity of cells correlate with the time-dependent contribution of the cell loss at initial stages of the treatment. An intermediary increase in the cellular reducing potential, or its delayed drop, may be observed due to the primary clearance of the most impaired cells in the population. At higher treatment doses (i.e., when either the agent concentration or treatment time are increased), a decrease in cellular NAD(P)H:XTT reductase activity summarizes the decreases in both the reducing potential and number of the affected cells.

Thus, a high concentration of extracellular ThDP reduces viability of A549 cells, testified by decreases in their NAD(P)H:XTT reductase activity and glutathione homeostasis (Figure 2). In the standard medium, A549^p21-^ cells handle the negative effect of ThDP supplementation better than A549^wt^ cells, which is associated with the ThDP-induced p53 expression in A549^p21-^ cells only. In contrast, the exposure to thiamine deficiency induces p53 only in A549^wt^ cells (Figure 2M–P), which predisposes such cells to handle the negative ThDP effect better than A549^p21-^ cells do (Figure 2 and Figure 4). The essential role of the OGDHC function in the regulation of the viability of A549 cells is demonstrated by the inverse correlation of the viability with the percentage of OGDHC holoenzyme (Figure 3C), which is controlled by the interdependent expression of p53 and p21 (Figure 1 and Figure 2).

### 2.4. Inhibitor of OGDHC, Succinyl Phosphonate, Interferes with the Changes in Cellular Redox State, Induced by the OGDHC Activator ThDP

In view of the negative correlation between cellular redox state and activity of holo-OGDHC (Figure 3), we have further applied a specific OGDHC inhibitor, succinyl phosphonate (SP) [1,2,5,6], to show that the paradoxical pro-oxidant effect of ThDP on A549 cells, which is increased by p21 knockdown, involves the coenzyme action of ThDP in the OGDHC-catalyzed reaction. Figure 4 demonstrates that under all conditions but the thiamine-deficiency of A549^wt^ cells, SP alleviated (*p* < 0.001) the pro-oxidant effect of ThDP, leading to an increase in cellular NADP(H) reductase activity, compared to that in the presence of ThDP alone. Only in the thiamine deficient A549^wt^ cells (Figure 4C,G), where ThDP increases the NAD(P)H:XTT reductase activity of A549^wt^ cells, does SP act the opposite way, exhibiting a strong (p <0.002) interaction with the ThDP antioxidant action (Figure 4C,G). A highly significant (*p* < 0.003) interaction between the action of SP and ThDP on the NAD(P)H:XTT reductase activity, is also revealed in the control A549^wt^ cells (Figure 4A,D). In contrast, A549^p21-^ cells do not demonstrate any significant interaction between the SP and ThDP effects on cellular NAD(P)H:XTT reductase activity in either the control or thiamine deficient state (Figure 4).

Thus, under all the conditions when the OGDHC activator ThDP does not increase cellular redox potential, the OGDHC inhibitor SP does (*p* < 0.001, Figure 4E,F,H). However, the positive effect of ThDP on cellular NAD(P)H:XTT reductase activity of the thiamine deficient A549^wt^ cells is abrogated by SP (Figure 4G). As a result, not only the OGDHC coenzyme ThDP, but also the OGDHC inhibitor SP demonstrate dual action on cellular reducing power in the p21-dependent manner.

It is remarkable, that the NAD(P)H:XTT-reducing activity of A549^p21-^ cells is increased by the OGDHC inhibitor SP in both the control and thiamine deficient state, whereas that of A549^wt^ cells shows a more complex behavior regarding the SP effect (Figure 4). This finding reciprocates the differences observed in the ThDP-treated A549^p21-^ and A549^wt^ cells regarding the perturbed glutathione homeostasis, p53 induction and OGDHC saturation with ThDP (Figure 1 and Figure 2). All the differences are especially pronounced after cellular exposure to thiamine deficiency, i.e., when A549^p21-^ and A549^wt^ cells strongly differ in their p53 expression (Table 1 and grey bars in Figure 2O vs Figure 2P, *p* = 0.04).

### 2.5. The p21-Dependent Anticancer Drug Cisplatin Has Similar and Non-Additive to Thiamine/ThDP Actions on Viability of A549 Cells

Because the action of 5 mM ThDP in A549 cells compromised cellular redox state in a p53-, p21- (Figure 2), and OGDHC-dependent (Figure 3) manner, we compared the ThDP effects to those of a well-studied anticancer drug cisplatin, whose toxicity is also known to impair cellular redox state and be linked to the p21 expression [38,39,40] and OGDHC function [41]. In good accord with these published results, we observed a five-fold higher toxicity of cisplatin in A549^wt^ cells (the white bar in Figure 5E), compared to A549^p21-^ cells (the white bar in Figure 5F). In our experiments with a varied availability of thiamine in the cell culture medium, we also observed conditional significance of p21 expression for the cisplatin toxicity, known from other studies [47,48,49]. In contrast to the cells in the standard medium, where the cisplatin toxicity in A549^p21-^ cells is less than in A549^wt^ cells (the white bars in Figure 5F vs Figure 5E), in the thiamine deficient medium, cisplatin affects A549^p21-^ cells more than A549^wt^ cells (the white bars in Figure 5H vs Figure 5G). However, this p21-dependent increase in the cisplatin toxicity in the thiamine deficient medium originates from the abrogation of the “proliferative” effect of cisplatin in the thiamine deficient A549^wt^ cells (the white bars in Figure 5G). As discussed above, this intermediary increase in the NAD(P)H:XTT reductase activity probably occurs due to the elimination of the most impaired cells at initial stage of the treatment. With a low concentration of ThDP (0.05 mM, light-grey bars in Figure 5) added to the thiamine deficient cells, the p21 dependence of the cisplatin effect approaches that in the standard medium: The statistically significant toxic effect of cisplatin in A549^wt^ cells becomes insignificant in A549^p21-^ cells (the light-grey bars in Figure 5G vs Figure 5H). Accordingly, p21 knockdown decreases the cisplatin effect on A549 cells under all conditions (Figure 5). However, the combinations of the thiamine deficient medium with cisplatin (the white bar in Figure 5G), or with the OGDHC inhibitor SP (Figure 4G) increase the reducing power of the A549^wt^ cells, presumably due to a better clearance of the impaired cells.

Analysis by the two factor ANOVA (Tables under the graphs in Figure 4 and Figure 5) indicates that A549^wt^ cells demonstrate the highest interaction between the effects of the OGDHC activator ThDP and the OGDHC inhibitors, either SP [1,2,5,6] (Figure 4) or cisplatin [41] (Figure 5), on cellular viability (*p*-values for the interaction < 0.003). This regulation of the ThDP effects on A549^wt^ cells by the OGDHC inhibitors provides further evidence on the mediation of the ThDP effect by OGDHC. The p21 knockdown strongly decreases the interaction, up to its complete disappearance (p-values 0.9 and 0.4 for the interaction in Figure 4F and Figure 5F) under the conditions when A549^p21-^ cells do not significantly change the OGDHC saturation by ThDP in response to ThDP (Figure 2, A549^p21-^ cells under the control conditions). Thus, the p21 knockdown in A549 cells decreases the impact of the known inhibitors of OGDHC, SP, or cisplatin on the changes in A549 viability induced by the OGDHC activator ThDP. The weakest regulation of the ThDP effects by SP or cisplatin (Figure 4and Figure 5B) coincides with the lowest response to 5 mM ThDP of the OGDHC partitioning into the active (holo) and latent (apo) forms (Figure 2, A549^p21-^ cells grown in the standard medium). Thus, the interacting effects of ThDP and SP or cisplatin on A549 cells (Figure 5) are mediated by OGDHC under the control of the p53–p21 axes.

The cisplatin effects are non-additive to those of ThDP (Figure 5). That is to say, cellular responses to cisplatin (the white bars in Figure 5E–H) are no more evident in the presence of 5 mM ThDP (the black bars in Figure 5E–H). As already noted above, in the thiamine deficient A549^wt^ cells cisplatin even increases the cellular reductase activity (Figure 5C, the white hatched bar vs the white bar), similar to that of a low (0.05 mM) concentration of ThDP (Figure 5C, the light-grey bar vs the white bar). In this specific case, the effects of cisplatin and ThDP are non-additive as, upon their combined action, cisplatin acts as an antagonist of ThDP, decreasing the proliferative action of ThDP added to the thiamine deficient A549^wt^ cells (Figure 5G, the light-grey bar vs the white bar). Overall, the data on the similar non-additive actions of ThDP and cisplatin under the variety of conditions used (Figure 5), point to the competitive action of ThDP and cisplatin in the same pathways involving regulatory interactions between the cellular thiamine pool, OGDHC, and p53–p21 axes.

### 2.6. Comparison of the ThDP Action in the Non-Cancer and Cancer Epihtelial Cell Lines

The pro-oxidant action of a high ThDP concentration on A549 cells is paradoxical from the conventional viewpoint on ThDP as a coenzyme of ThDP-dependent enzymes, participating in cellular NAD(P)H production in any cell type. Hence, the malignant (A549) and normal (Vero) epithelial cell lines were compared regarding the studied ThDP effects. The comparison (Figure 6) indicates that the ThDP-induced decreases in cellular reducing power (Figure 6A) and glutathione (Figure 6B) are specific to A549 cells and not observed in Vero cells. Once again, this difference is linked to the cell-specific responses of OGDHC to a high ThDP concentration: The partitioning of cellular OGDHC between the active (holoenzyme) and latent (apoenzyme) forms in the A549 and Vero cells undergoes opposite changes in response to ThDP (Figure 6C,D). Although 5 mM ThDP does not significantly change the total OGDHC activity levels in either A549 or Vero cells (Figure 6E), the normal epithelium tends to decrease its holo-OGDHC level in the presence of 5 mM ThDP (Figure 6C), whereas the adenocarcinoma epithelium exhibits a five-fold increase in holo-OGDHC under the same conditions (Figure 6D). It may thus be concluded that cellular mechanisms controlling the formation of holo-OGDHC differ in the Vero and A549 epithelial cells.

It is worth noting that the opposite reactivities of OGDHC and cellular reducing power to 5 mM ThDP in the A549 and Vero cells are observed at very different initial levels of the studied parameters. As shown in Table 2, the holo-OGDHC level is 24-times higher in the Vero vs A549 cells, with 96% of total OGDHC present in its active holoform in Vero cells. In contrast, in A549 cells up to 76% of OGDHC is in the latent apoform (Figure 6D). Thus, a 30-fold higher glutathione level corresponds to a 24-fold lower holo-OGDHC level in the A549 vs Vero cells. Our data on the striking difference in the glutathione content in the normal and cancer epithelium (Table 2) agree with independent quantifications, which have shown an order of magnitude higher glutathione content in A549 cells, compared to the lung fibroblast cell lines [50,51]. All the data are consistent with the general notion that high levels of glutathione are observed in rapidly proliferating cells, where glutathione also participates in nuclear processes and cell cycle regulation [52]. The observed glutathione content and GSSG/GSH ratio (Figure 2, Table 2) agree well with the values reported for the A549 [51,53,54,55,56] and Vero cells [57,58]. Similar levels of cellular OGDHC activities are also reported in independent studies of A549 cells [6], although the partitioning into the holo- and apoforms has been evaluated only in the present work.

Thus, a comparison of the normal (Vero) and adenocarcinoma (A549) epithelial cells (Table 2) corroborates the inverse relationship between the cellular reducing potential and OGDHC holoenzyme level in A549 cells (Figure 3). The glutathione level is high in adenocarcinoma A549 cells with a low level of the OGDHC holoenzyme. Vice versa, the glutathione level is low in normal epithelial cells with a high level of the OGDHC holoenzyme (Table 2).

## 3. Discussion

A medically important contribution of the present work is the demonstration of a strong negative effect of a high dose of ThDP (known in medicine as cocarboxylase) on the viability of the adenocarcinoma epithelial cell line A549, which is not observed in a non-cancer epithelium cell line (Figure 6), nor acknowledged in any previous studies on the effects of ThDP and its precursor thiamine in a number of chemical [59,60] and biological systems [33,34,35,36,37]. After the incubation with 5 mM ThDP for 24 h, the viability impairment in A549 cells is manifested as a 70% decrease in cellular NAD(P)H:XTT reductase activity and 20–30% lower levels of cellular glutathione (Figure 6), accompanied by a 30% reduction of the glutathione redox potential (GSH/GSSG, Table 1) in A549^wt^ cells. In a recent study, the tumorigenic (MCF7) and non-tumorigenic (MCF10A) breast epithelial cell lines [32] demonstrated a similar difference in their response to thiamine (vitamin B1, the cocarboxylase precursor): The addition of thiamine to cell culture media decreased cellular NAD(P)H:MTT reductase of MCF7 cells by 63%, not affecting the viability of MCF10A cells. Thus, the two independent studies using either thiamine [32] or its major cellular derivative ThDP (current work) reveal no negative action of thiamine or ThDP on normal metabolism of different cell types in the concentration range where the compounds are damaging to the A549 and MCF7 cancer cells. During the last two decades, the negative action of high doses of thiamine has been sporadically observed also in other studies of the thiamin supplementation to different types of cancer cells [29,30,31]. The negative action of ThDP, which is employed in our study, has not been examined, because ThDP has not been supposed to penetrate cellular membranes. However, according to a recent study, ThDP enters cells through a specific transporter SLC44A4 [61] not known before, and this transporter is well expressed in epithelial cells. In this regard, ThDP action on cancer epithelial cells may be more effective, than that of thiamine, because the former does not require the intracellular phosphorylation of the latter by thiamine diphosphokinase.

Our mechanistic studies show for the first time the link between the antiproliferative ThDP action on A549 cells and the p53–p21 pathway, pointing out that the p21 knockdown promotes the deteriorating effect of ThDP on cellular glutathione and NAD(P)H pools (Figure 2). A 20% decrease in cellular protein, caused by ThDP supplemented to the thiamine deficient A549^wt^ cells, is not observed in A549^p21-^ cells (Table 1), suggesting that the thiamine deficiency pre-disposes only A549^wt^ cells to a more efficient cell sorting under the ThDP-caused damage. The pre-disposition obviously depends on the increased p53 expression in the thiamine deficient A549^wt^ cells, not occurring in the cells with p21 knockdown (Figure 2). The pre-disposed thiamine deficient A549^wt^ cells seem to better eliminate the cells whose reducing power is impaired not only by ThDP, but also by SP or cisplatin (Figure 4 and Figure 5). Thus, intact signaling through the p53–p21 pathway supports the efficient elimination of the OGDHC-mediated cell damage. This finding is in good accord with the recently established role of the bimodal switch in the p53 action in A549 cells [62,63] and the known duality in the p53 action in cell survival [64]: The p53-provided DNA damage repair requires the cell cycle arrest supported by the p53 oscillations, whereas the monotonous accumulation of the damage and p53 causes the p53-induced cell death. Accordingly, the p21 knockdown exacerbates the negative effect of ThDP on the viability of A549 cells (Figure 2), because a poor ability of A549^p21-^ cells to undergo the cell-cycle arrest is incompatible with the p53-dependent damage repair and efficient cells sorting.

Finally, we show that the negative p21-dependent effects of ThDP on the reducing potential of A549 cells are mediated by a key ThDP-dependent complex of the tricarboxylic acid cycle in mitochondria, OGDHC. The highly significant negative correlation between the viability of A549 cells and the percentage of holo-OGDHC is shown (Figure 3C). The inverse relationship is independently corroborated by the comparison of the levels of glutathione and holo-OGDHC in the cancer (A549) and normal (Vero) epithelium (Table 2), and a strong decrease in cellular viability and glutathione when the holo-OGDHC level is increased in response to 5 mM ThDP in A549^wt^ cells only (Figure 6). Moreover, in A549^wt^ cells ThDP increases oxidative stress, as the decreased reducing potential of cellular glutathione pool shows (GSH/GSSG ratio, Figure 2I,K). In view of the well-known ability of holo-OGDHC to catalyze side reaction of ROS production under conditions of perturbed homeostasis [1,2,3], a role of this side reaction in the ThDP effects in A549 cells may be suggested. Using the OGDHC specific inhibitor SP confirms the suggestion, as SP interferes with the action of ThDP (Figure 4). The suggestion is further corroborated by the competitive action of ThDP and cisplatin (Figure 5), as the impairment of cellular redox state is a known mechanism of the cisplatin action [65,66]. Thus, both the attenuation of the ThDP decrease of cellular viability by the OGDHC inhibitor SP and the non-additivity of the effects of ThDP and a ROS inducer cisplatin (Figure 4 and Figure 5) support the assumption that “over-activation” of OGDHC by ThDP is damaging for A549 cells because of the enzyme-catalyzed side reaction of ROS generation. This mechanism of the pro-oxidant action of ThDP agrees with independent results on the association of the up-regulated *OGDHL* gene expression and increased ROS levels, considered in Section 1 [5,8,9,10,11], and independent experimental evidence that OGDHC possesses the highest capacity to produce ROS compared to other family members [67].

It should also be taken into account that the OGDHC-catalysed reaction has an important role in the metabolism of glutamate [68]. Intracellular glutamate is not only a precursor for the glutathione biosynthesis itself, but also participates in the cellular exchange for extracellular cystine. The exchange has special significance in cancer cells which often overexpress the corresponding transporter SLC7A11 to increase the intracellular transport of cystine for glutathione biosynthesis ([5] and references therein). The link between the decreased OGDHC activity in cancer vs normal epithelium and the higher glutathione level in the former vs latter cells (Table 2) is further supported by our finding that, in the cancer-specific metabolism, intracellular concentrations of glutathione may be increased by thiamine deficiency (Figure 2). Because the deficiency decreases the OGDHC activity (Figure 1), the glutathione biosynthetic precursors glutamate and cysteine may be increased according to the mechanisms discussed above. Our data indicate that the OGDHC association with the glutathione biosynthesis is stronger in A549^p21-^ vs A549^wt^ cells. Contrary, the ROS-producing activity of OGDHC contributes to the glutathione homeostasis in A549^wt^ more than in A549^p21-^. Such differences are associated with the different reactivity of p53 to the thiamine deficiency and ThDP supplementation in the A549^wt^ and A549^p21-^ cells (Figure 2), supported by independent data on the role of p53–p21 axes in glutathione metabolism of cancer cells [45].

Glutathione is known to protect OGDHC from the self-inactivation, which otherwise limits the side reaction of ROS production (reviewed in [1,2,3]). Hence, a high glutathione content, observed in A549 cells (Table 2), should increase the pro-oxidant action of ThDP, mediated by holo-OGDHC ROS production, compared to the normal epithelium with a 30-fold lower glutathione level. The corresponding minimization of the holo-OGDHC formation in A549 cells under basal conditions, where 76% of the enzyme is in its latent apoform, may manifest an adaptation to limit ROS under the high glutathione content in the A549 cells (Table 2). Overexpression of glutaredoxin and thioredoxin in cancer cells [69] may further perturb the mechanisms limiting production of ROS by OGDHC in normal cells. Like glutathione, thioredoxin protects OGDHC from self-inactivation upon perturbed flux through OGDHC, increasing ROS production by OGDHC [2,3]. Glutaredoxin activates OGDHC by deglutathionylation, which may increase both the physiological and ROS-producing function [3]. Conditional outcome of this regulation of OGDHC may underlie the observations that glutaredoxin overexpression may disrupt both apoptotic and survival signaling pathways [70], similar to the dual action of the OGDHC activator ThDP and the OGDHC inhibitors SP or cisplatin in our study (Figure 4 and Figure 5), or to the dual action of p53 [62,63,64]. Thus, such known features of cancer cells as increased glutathione and the up-regulated thiol-disulfide reductases thioredoxin and glutaredoxin, may promote the ROS production by OGDHC, requiring decreased levels of holo-OGDHC in cancer vs normal cells (Table 2). The high sensitivity of cancer cells to inactivation of glutathione-dependent lipid peroxidases [71] may provide another link to the OGDHC-generated radical species through the radicals of the complex-bound lipoate residues. Generation of these OGDHC-bound radical species is shown to occur also under anaerobic conditions [72], and may affect the OGDHC-interacting membrane compartment [73]. Because this mechanism of radical formation does not require oxygen, its significance may be increased by hypoxic conditions in cancer cells, simultaneously increasing the role of the glutathione-dependent lipid peroxidases in such cells [71].

Thus, molecular mechanisms of the mediation of ThDP effects on A549 cells involve the OGDHC partitioning between the active holoenzyme (with ThDP bound) and latent apo-enzyme (potentially activated by ThDP binding). The partitioning is deregulated by p21 knockdown in such a way that A549^p21-^ cells cannot adjust the OGDHC holoenzyme formation to the level of total OGDHC, as A549^wt^ cells do. This difference coincides with the different reactivity of p53 to the changed thiamine status in A549^wt^ and A549^p21-^ cells (Figure 2).

## 4. Materials and Methods

### 4.1. Cell Cultivation and Reagents

Continuous cell lines of primate epithelium were used in the study: Human epithelial adenocarcinoma non-small cell lung cancer cell line A549 (ATCC® CCL-185™) and green monkey (*Cercopithecus Aethiops*) renal ducts epithelium cells Vero (RCCC, St-Petersburg, RF). The malignant and normal cells were cultured under the same conditions, i.e., in Dulbecco’s Modified Eagle’s medium (DMEM) (Thermo Fischer Scientific, 21855), which comprised 1g/L glucose, 1 mM pyruvate, 4mM GlutaMAX™, 10% FBS (Gibco, 10270, E.U. Approved (South American)), 100 U/mL penicillin, and 0.1 mg/mL streptomycin (Thermo Fisher Scientific, 15140, Bleiswijk, Netherlands) at 37 °C with 5% CO_2_ in a humidified atmosphere. Supplementation of the cells with ThDP (0–5 mM) (Sigma, C8754, Gillingham, UK), succinyl phosphonate (SP, 4.5 mM) (synthesized according to [74], with its identity and purity proven by NMR spectroscopy), and/or cisplatin (5 μM) was done by exchanging the media to the fresh one comprising the studied compounds 24 h prior to the viability assays or cells lysis.

Puro siRNA expression vector *pSilencer*™ 2.1-U6 (Thermo Fisher Scientific, AM5762) and TurboFect Transfection Reagent (Thermo Fisher Scientific, R0531) and RNeasy mini kit (Qiagen, 74106, Hilden, Germany) were used. The protease and phosphatase inhibitor cocktails cOmplete™ (Roche, 04693116001, Mannheim, Germany) and PhosSTOP™ (Roche, PHOSS-RO, Mannheim, Germany) were used. NAD^+^ (N7004), 2-oxoglutarate (K1750), and CoA (C4282) were purchased from Sigma.

### 4.2. Cellular Model of Thiamine Deficiency

Home-made DMEM analogous to DMEM used for cell growth, was prepared for experiments involving thiamine deficiency. The thiamine-free DMEM medium contained dialyzed 10% FBS (Sigma, F0392), 4mM L-Glutamine, 1g/L glucose, 1 mM pyruvate, and no added thiamine. The same home-made DMEM with 4mg/L thiamine hydrochloride was used in the experiments with the control cells, to which the thiamine deficient cells were compared. Establishment of the thiamine deficient A549 cells was made by their plating into the thiamine-free DMEM, followed by seven days growth in the thiamine-free medium prior to experiments. During this time, the medium was exchanged twice, i.e., after the overnight attachment of the cells and on the fourth day.

### 4.3. Construction of Stable p21 Knockdown A549 Cell Line

The targeting sequence p21 is: 5’-GCGATGGAACTTCGACTTT-3’. The shRNA expression plasmid was constructed according to the manufacturer’s instruction. Briefly, two specific oligonucleotides were designed using online tools BLOCK-iT™ RNAi designer (http://rnaidesigner.lifetechnologies.com/rnaiexpress/). The sequences are: 5’-GATCCGCGATGGAACTTCGACTTTTTCAAGAGAAAAGTCGAAGTTCCATCGCTTTTTTGGAAA-3’ and 5’-AGCTTTTCCAAAAAAGCGATGGAACTTCGACTTTTCTCTTGAAAAAGTCGAAGTTCCATCGC-3’. The two complimentary oligonucleotides were annealed and ligated into pSilencer 2.1-U6 puro siRNA expression vector (Thermo Fisher Scientific, AM5762) and transfected into A549 cells with TurboFect Transfection Reagent (Thermo Fisher Scientific, R0531). A non-targeting negative control plasmid from the kit was used as a mock control. Stably expressing cells were selected with puromycin (Thermo Fisher Scientific, Gibco™ A1113803) at 1 μg/mL for two weeks and the knock-down clones were determined with anti-p21 antibody (Abcam, ab188224). The beta-actin was used as loading control (Sigma, A5441). The knock-down and mock control cells were maintained with puromycin at concentration of 0.2 μg/mL.

A549^p21-^ cells used in our work possess a much lower level of p21 protein (Appendix A). As a result, the A549^p21-^ cells increase their proliferation rate, demonstrating smaller cell size, compared to A549^wt^ cells (Appendix A) as determined by a mini automated cell counter (Moxi, Orflo Technologies). Such effects are in accordance with p21 function as a cyclin-dependent kinase inhibitor, whose knockdown would accelerate the cell cycle. Among other effects of the p21 knockdown on cell morphology, increased formation of colony islands in A549^p21-^ vs A549^wt^ cells has been observed (Appendix A). 

### 4.4. Cell Viability Assay

Cellular viability was determined with a Cell Proliferation Kit II (Roche, 11465015001) according to the manufacturer’s instruction. Briefly, 50 μL of XTT labeling reagent and 1 μL of electron coupling reagent were mixed, and 50 μL of the mixture was added into each well containing 100 μL of culture medium. The plate was incubated at 37 °C with 5% CO_2_ in a humidified atmosphere for 2 h. The absorbance was then measured at 450 nm with a reference wavelength at 650 nm. The wells without cells, with the same volume of the control and experimental culture media were used as the blanks. The influence of a treatment was determined as percentage of the corresponding control without the treatment. Due to the relative nature of the assay, the comparison of cellular viabilities under different conditions was limited to one cell type only.

### 4.5. Preparation of Cell Lysates 

Briefly, A549 cells were seeded by 2 × 10^5^ cells in each well of 6-well plates. After 24 h the medium was replaced by the same medium supplemented with ThDP or succinyl phosphonate (SP). The cells were cultured for 24 h (approximately to 70% confluence). They were then washed with PBS twice, and lysed (30 min shaking on ice) in radioimmune precipitation assay buffer (50 mM HEPES, pH 7.5, 150 mM NaCl, 1% Nonidet P-40, 0.05% sodium deoxycholate, and the protease and phosphatase inhibitor cocktails). The lysed cells were scraped and the lysates collected. 

### 4.6. OGDHC Activity Assays

Enzyme activity of OGDHC was assayed in cell lysates at the day of preparation under conditions described previously [75]. The reaction medium to assay OGDHC with tightly bound ThDP, i.e., endogenous level of the OGDHC holoenzyme, included 20 mM potassium-phosphate buffer, pH 7.0, containing 2.5 mM NAD^+^, 1 mM dithiothreitol, 1 mM CaCl_2_, 1 mM MgCl_2_, 0.1 mM CoA, and 2 mM 2-oxoglutarate. The same medium omitting 2-oxoglutarate was used for assaying the blank reaction rate to subtract from the 2-oxoglutarate-dependent reaction rates. The lysates were assayed at two different protein concentrations, each concentration in quadruplicate. The linear part of the product accumulation curves was used for the reaction rate determination. The reaction rate was at least 4 times higher than the background. To determine total cellular OGDHC activity, which characterizes expression of the functional complex, 1 mM ThDP was added to the assay medium. Cellular level of the OGDHC apoenzyme was calculated from the difference between the total and holo-OGDHC activities. Partition of the total OGDHC into the holo- and apoenzymes was characterized by the % ratio of holo-OGDHC to total OGDHC.

### 4.7. Assay of p53 by Western Blotting

The levels of p53 protein were assayed using anti-p53 (Santa Cruz Biotechnology, sc-126) primary antibodies. The anti-β-actin (Sigma, A5441) and anti-VDAC/porin (Cell Signaling Technology, 4661) antibodies were used as loading controls, with the p53 protein levels normalized to the sum of the loading control levels (Appendix A). To compare the p53 protein levels under different conditions, the percentage of the normalized p53 protein in A549^wt^ cells was used.

### 4.8. Preparation of Methanol–Acetic Acid Extracts

Cellular extracts were obtained according to established protocol [76]. Briefly, cells on 6-well plates were put on ice, washed with PBS twice, and extracted with cold water solution of 40% methanol with 0.12% acetic acid. The plates were slowly shaken for 30 min. Cells were scraped, collected in 1.5 mL tubes and centrifuged at 6200× *g* for 15 min at 4 °C. The supernatants for glutathione (GSH) assays and pellets for total protein assays were separated and stored at −70 °C. 

### 4.9. Glutathione Assay

Briefly, 5,5′-dithiobis-2-nitrobenzoic acid (DTNB) was used for the determination of free glutathione (GSH) in the methanol-acetic acid extracts as recently described [33]. The method of Hissin and Hilf [77] was used for glutathione disulfide determination with o-phthalaldehyde. The cellular content of glutathione and its disulfide were normalized per mg of protein in the extracts.

### 4.10. Total Protein Assay

The total protein content was determined using the Bio-Rad Protein Assay Kit I, 5000001, according to manufacturer’s instructions.

### 4.11. Statistical Analysis

Values are presented as mean ± SEM. Independent experiments refer to different cell batches. Statistical analysis was performed using GraphPad Prism, version 7.0 (GraphPad Software Inc., La Jolla CA, USA). Comparisons between multiple experimental groups were done using two-way ANOVA with post-hoc Tukey’s test. Differences with *p* < 0.05 were considered significant. Differences with 0.05 < *p* < 0.1 were considered as trends.

The Gaussian distribution of the parameters was shown by the D’Agostino & Pearson omnibus normality test, enabling correlation analysis using Pearson criterion. In addition, we performed the recommended comparison of the Spearman and Pearson correlations [78] for the investigated data samples, which did not reveal significant differences between the two types of the analyses. 

## 5. Conclusions

In summary, our study has demonstrated the OGDHC-influenced interplay between the p53–p21 axes and multimodal metabolic regulators ThDP and cisplatin. The investigated mechanisms controlling the interactions of such regulators point to a promising new direction in the development of combinatorial therapies and fight against the drug resistance problem.

## Figures and Tables

**Figure 1 ijms-21-03759-f001:**
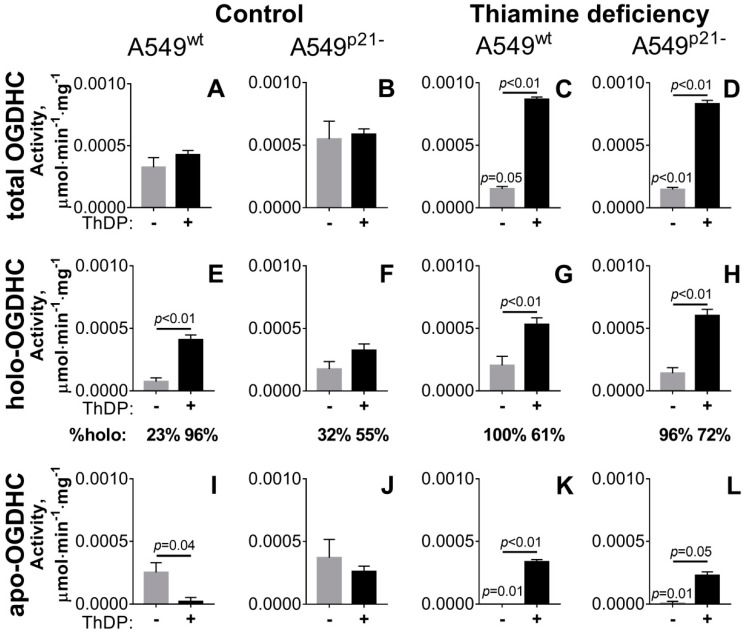
p21-Dependent changes in the total activity of OGDHC (**A**–**D**) and its partition into the holo- (**E**,**F**) and apoenzymes (**I**–**L**) at the exposure of A549 cells to varied thiamine/ThDP levels. A549^wt^ or A549^p21-^ cells were cultured in DMEM prepared either with 4 mg/L thiamine hydrochloride (Control, **A**,**B**,**E**,**F**,**I**,**J**) or without thiamine (Thiamine deficiency, **C**,**D**,**G**,**H**,**K**,**L**). Analysis was made 24 h after the media were exchanged for those without (gray columns) or with 5 mM ThDP (black columns). The OGDHC partition into the apo- and holoenzymes was determined as described in the text and Materials and Methods. Under the graphs E-H the OGDHC holoenzyme activities are shown as % of total OGDHC activities (“%holo”). The data of at least three independent experiments are presented as mean ± SEM. Statistical significance of the differences between all the groups (shown in full in Table 1) was determined by two-way ANOVA followed by Tukey’s post hoc test. On the graphs, only the differences due to the ThDP/thiamine variations are indicated. The *p*-values above the horizontal lines or the grey bars characterize the changes due to the ThDP addition or thiamine deficiency, correspondingly.

**Figure 2 ijms-21-03759-f002:**
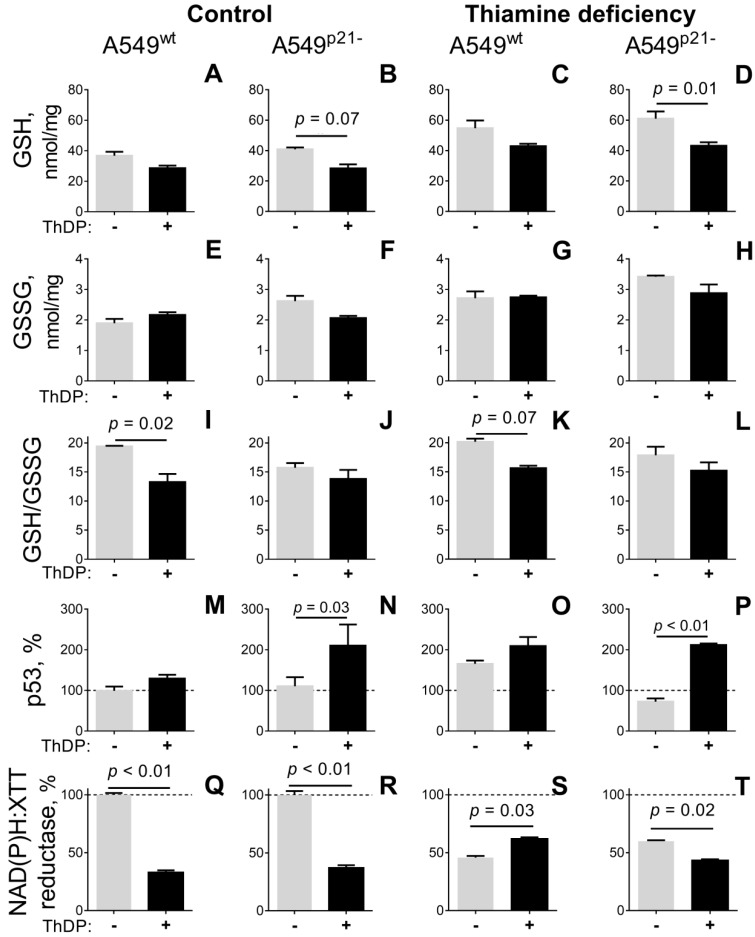
Influence of p21 knockdown on the thiamine/ThDP-induced changes in the levels of glutathione (**A**–**D**), glutathione disulfide (**E**–**H**), their ratio (**I**–**L**), p53 protein level (**M**–**P**) and NAD(P)H:XTT reductase activity (**Q**–**T**) of A549^wt^ and A549^p21-^ cells. The cells were cultured in DMEM prepared with 4 mg/L thiamine hydrochloride (Control, A, B, E, F, I, J, M, N, Q, R) or without thiamine (Thiamine deficiency, C, D, G, H, K, L, O, P, S, T). After 24 h, the media were exchanged for those without (gray columns) or with 5 mM ThDP (black columns), and the indicated parameters were assayed as described in Materials and Methods. The data of at least three independent experiments are presented as means ±SEM. On the graphs, only the statistically significant differences of the ThDP effects are indicated by *p*-values above the horizontal lines connecting the compared groups. Statistical significances of the differences between all the groups are shown in Table 1. GSH—glutathione, GSSG—glutathione disulfide.

**Figure 3 ijms-21-03759-f003:**
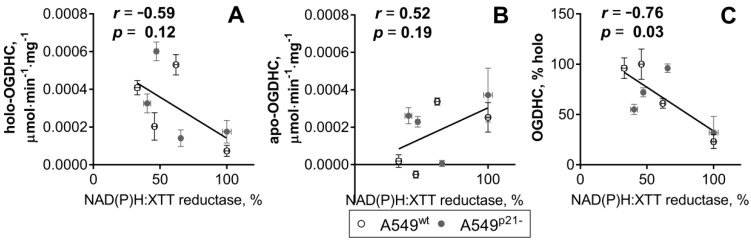
Correlations between the cellular NAD(P)H:XTT reductase activity and the OGDHC partition into the holo- and apo-OGDHC. The data for eight experimental groups including A549^wt^ and A549^p21-^ cells under different thiamine/ThDP saturation are used. All datasets successfully passed the D’Agostino & Pearson omnibus normality test. Linear regressions of Pearson’s correlation are presented, with their correlation coefficients (*r*) and p-values (*p*) for statistical significance given on the graphs. (**A**) Correlation between the NAD(P)H:XTT reductase activity and the holo-OGDHC level. (**B**) Correlation between the NAD(P)H:XTT reductase activity and the apo-OGDHC level. (**C**) Correlation between the NAD(P)H:XTT reductase activity and ratio (in %) of holo-OGDHC to total OGDHC.

**Figure 4 ijms-21-03759-f004:**
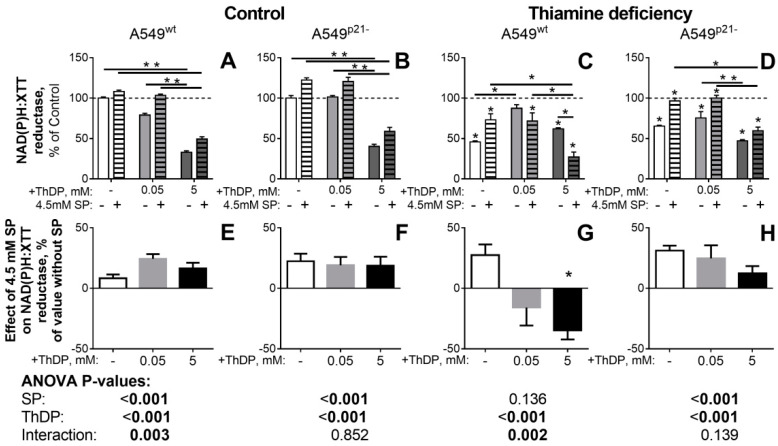
Influence of the OGDHC inhibitor SP on p21-dependent responses of NAD(P)H:XTT reductase activity of A549 cells to variation in the thiamine/ThDP levels. A549^wt^ (**A**,**C**,**E**,**G**) or A549^p21-^ (**B**,**D**,**F**,**H**) cells were cultured in DMEM prepared with 4 mg/L thiamine hydrochloride (Control, A, B, E, F) or without thiamine (Thiamine deficiency, C, D, G, H). 24 h after supplementation with indicated concentrations of ThDP (shades of grey) and/or SP (4.5 mM, hatched columns), NAD(P)H:XTT reductase activity was assayed. Values obtained without ThDP supplementation are presented by white columns, those at 0.05 or 5 mM ThDP—by gray or black columns, correspondingly. In the upper part (A–D), cellular NAD(P)H:XTT reductase activity is normalized to the activity of the control A549^wt^ or A549^p21-^ cells to show the combined effects of thiamine/ThDP and SP on cellular viability. In the lower part (E–H), cellular NAD(P)H:XTT reductase activity is normalized to the corresponding value without SP to show the specific effect of SP under each the varied thiamine/ThDP levels. The data of at least three independent experiments are presented as mean ± SEM. Statistically significant differences (*p* < 0.05, shown by asterisks) are determined by two-way ANOVA followed by Tukey’s post hoc test. Horizontal lines show the groups compared. Asterisks above the bars in C and D indicate the statistically significant differences between the corresponding thiamine deficient cells and cells with the standard thiamine level. Asterisk in G indicates the statistically significant effect of SP vs the same condition without SP. ANOVA significances of the factors (SP and ThDP) and their interactions are given below the graphs, with significant (*p* < 0.05) values in bold.

**Figure 5 ijms-21-03759-f005:**
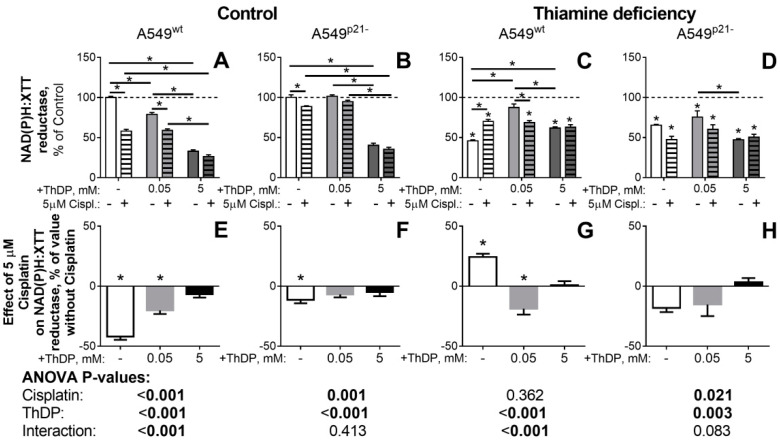
Influence of cisplatin (Cispl.) on p21-dependent responses of NAD(P)H:XTT reductase activity of A549 cells to variation in the thiamine/ThDP levels. A549^wt^ (**A**,**C**, **E**,**G**) or A549^p21-^ (**B**,**D**,**F**,**H**) cells were cultured in DMEM prepared with 4 mg/L thiamine hydrochloride (Control, A, B, E, F) or without thiamine (Thiamine deficiency, C, D, G, H). 24 h after supplementation with indicated concentrations of ThDP (shades of grey) and/or cisplatin (5 μM, hatched columns), NAD(P)H:XTT reductase activity was assayed. Values obtained without ThDP supplementation are presented by white columns, those at 0.05 or 5 mM ThDP—by gray or black columns, correspondingly. In the upper part (A–D), cellular NAD(P)H:XTT reductase activity is normalized to the activity of the control A549^wt^ or A549^p21-^ cells to show combined effects of thiamine/ThDP and cisplatin on cellular viability. In the lower part (E-H), cellular NAD(P)H:XTT reductase activity is normalized to the corresponding value without cisplatin to show the specific effect of cisplatin under each variation in the thiamine/ThDP levels. The data of at least three independent experiments are presented as mean ± SEM. Statistically significant differences (*p* < 0.05, shown by asterisks) are determined by two-way ANOVA followed by Tukey’s post hoc test. Horizontal lines show the groups compared. Asterisks above the bars in C and D indicate the statistically significant differences between the corresponding thiamine deficient cells and cells with standard thiamine level. Asterisks in E–G indicate the statistically significant effects of cisplatin vs the same condition without cisplatin. ANOVA significances of the factors (cisplatin and ThDP) and their interactions are given below the graphs, with significant (*p* < 0.05) values in bold.

**Figure 6 ijms-21-03759-f006:**
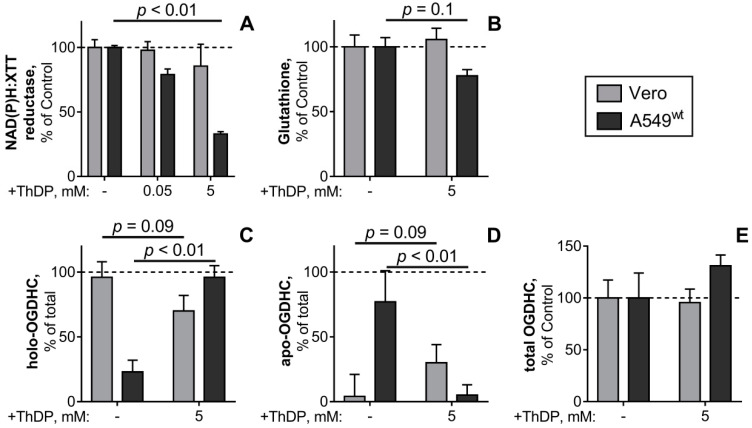
The ThDP-induced changes in the levels of NAD(P)H:XTT reductase activity (**A**), glutathione (**B**), OGDHC partition into the holo- (**C**) and apoenzymes (**D**) and total activity of OGDHC (**E**) of the cell lines Vero (grey bars) or A549^wt^ (black bars). The cells were cultured in standard DMEM, which was exchanged for the media with the indicated ThDP concentration (0–5 mM) 24 h before the assay of cellular NAD(P)H:XTT reductase activity or cell lysis for the glutathione and OGDHC assays. The data of at least three independent experiments are presented as means ±SEM. In A, B and D the % changes in cellular parameters are calculated relative to the values inherent in the same cells (A549^wt^ or Vero) under control conditions (100%). The absolute values for the glutathione levels and OGDHC activities, taken as 100%, are given in Table 2. In C and D, the percentage of the holo- and apo-OGDHC is calculated relative to the total OGDHC activity in each of the settings, as described in the text regarding the data in Figure 1. Statistical significance of the changes in a cell line was determined by one-way ANOVA followed by Tukey’s post hoc test for multiple groups (A) or by Mann-Whitney test for two groups (B-E), with p-values presented on the graphs.

**Table 1 ijms-21-03759-t001:** Effect of 24 h incubation with 5 mM ThDP on the assayed metabolic parameters of A549^wt^ and A549^p21-^ cells in the standard or thiamine-free DMEM. The data of at least three independent experiments are presented as means ±SEM. The eight experimental groups are marked by the letters (a–h) in the “Cell line” row. Statistically significant differences (*p* < 0.05) are determined by two-way ANOVA followed by Tukey’s post hoc test and are indicated as the difference of considered parameter to that in the corresponding experimental groups (a–h). Because the reaction rates of cellular NAD(P)H:XTT reductase are relative to the corresponding control cells, the comparison of this parameter is done only for different conditions applied to either A549^wt^ or A549^p21-^ cells. GSH—glutathione, GSSG—glutathione disulfide.

Medium	Standard DMEM				Thiamine-free DMEM			
Addition of ThDP for 24h	-		+ 5 mM ThDP		-		+ 5 mM ThDP	
Cell Line	A549^wt^ _a_	A549^p21-^ _b_	A549^wt^ _c_	A549^p21-^ _d_	A549^wt^ _e_	A549^p21-^ _f_	A549^wt^ _g_	A549^p21-^ _h_
**Protein, mg**	1.27 ± 0.03	1.14 ± 0.07 _e,h_	1.08 ± 0.05 _e,f,h_	1.13 ± 0.02 _e,h_	1.46 ± 0.04 _b,c,d,g_	1.38 ± 0.07 _c_	1.17 ± 0.05 _e,h_	1.44 ± 0.07 _b,c,d,g_
**Activity, μmol/(min∙mg)**								
**total OGDHC∙10^4^**	3.3 ± 0.8 _e,g,h_	5.5 ± 1.4 _e,f,g_	4.3 ± 0.3 _e,f,g,h_	5.9 ± 0.4 _e,f,g_	1.5 ± 0.7 _a,b,c,d,g,h_	1.5 ± 0.2 _b,c,d,g,h_	8.7 ± 0.2 _a,b,c,d,e,f_	8.3 ± 0.3 _a,c,e,f_
**apo-OGDHC∙10^4^**	2.5 ± 0.8 _c,e_	3.7 ± 1.4 _c,e,f_	0.2 ± 0.3 _a,b,g_	2.6 ± 0.4 _e,f_	0 ± 0.2 _a,b,d,g,h_	0.1 ± 0.2 _b,d,g,h_	3.4 ± 0.2 _c,e,f_	2.3 ± 0.3 _e,f_
**holo-OGDHC∙10^4^**	0.7 ± 0.3 _c,d,g,h_	1.8 ± 0.6 _c,g,h_	4.1 ± 0.4 _a,b,f_	3.3 ± 0.5 _a,h_	2.0 ± 0.3 _g,h_	1.4 ± 0.4 _c,g,h_	5.3 ± 0.5 _a,b,e,f_	6.0 ± 0.5 _a,b,d,e,f_
**GSH, nmol/mg**	38 ± 3 _e,f_	41 ± 1 _f_	29 ± 2 _e,f_	28 ± 3 _e,f,h_	55 ± 5 _a,c,d_	61 ± 5 _a,b,c,d,g,h_	43 ± 2 _f_	43 ± 2 _d,f_
**GSSG, nmol/mg**	1.9 ± 0.1 _e,f,g,h_	2.6 ± 0.2 _f_	2.2 ± 0.1 _f_	2.1 ± 0.1 _f,h_	2.8 ± 0.2 _a_	3.4 ± 0.1 _a,b,c,d_	2.7 ± 0.1 _a_	2.9 ± 0.3 _a,d_
**GSH/GSSG**	19.4 ± 0.1 _c,d_	15.7 ± 0.8	13.3 ± 1.4 _a,e_	13.8 ± 1.6 _a,e_	20.2 ± 0.5 _c,d_	17.9 ± 1.5	15.6 ± 0.4	15.2 ± 1.4
**p53, %**	100 ± 10 _d,g,h_	110 ± 22 _d,g,h_	129 ± 9	210 ± 52 _a,b,f_	166 ± 8 _f_	72 ± 8 _d,e,g,h_	209 ± 22 _a,b,f_	212 ± 4 _a,b,f_
**NAD(P)H:XTT reductase, %**	100 ± 2 _c,e,g_	100 ± 3 _d,f,h_	33 ± 2 _a,g_	40 ± 3 _b_	46 ± 1 _a_	62 ± 1 _b_	65 ± 1 _a,c_	47 ± 1 _b_

**Table 2 ijms-21-03759-t002:** Inverse relationship between the OGDHC activity and glutathione level, assayed in the Vero and A549^wt^ cell lines under control conditions. The data of at least three independent experiments are presented as means ± SEM.

Cell Line	Vero	A549^wt^
**GSH, nmol/mg**	1.2 ± 0.1	37.7 ± 2.6
**Activity, nmol/(min∙mg)**		
**total OGDHC**	1.76 ± 0.30	0.33 ± 0.08
**holo-OGDHC**	1.68 ± 0.22	0.07 ± 0.03
**apo-OGDHC**	0.07 ± 0.30	0.25 ± 0.08

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
