# Peer review of "Activation of Mitochondrial 2-Oxoglutarate Dehydrogenase by Cocarboxylase in Human Lung Adenocarcinoma Cells A549 Is p53/p21-Dependent and Impairs Cellular Redox State, Mimicking the Cisplatin Action"

_ijms, 2020, doi:10.3390/ijms21113759_

Round 1

Reviewer 1 Report

This study is not acceptable in its present form. The style of writing is poor and impossible to understand and much too much tekst is used. See for example the Results section: the text needs 7 pages, without Figures, Tables and Legends. It should be done in 2 pages maximum. Now it is impossible to understand what the authors are trying to prove.

In fact, only the first paragraph of the Discussion is well written.

Furthermore, only one cancer cell line (human lung cancer) and one non-cancer cell line (monkey kidney cell line) have been used. First, applying only one cell line is not done anymore already for a long time, because it cannot be proven whether the findings are cancer-related or specific cell line related. Furthermore a monkey kidney ductular epithelium is not at all a good control for human lung cancinoma cell line.

Finally, a clear aim of the study and clear conclusions are not described (see the Abstract, for example).

Reviewer 2 Report

Authors describe that activation of mitochondrial 2-oxoglutarate dehydrogenase by cocarboxylase in human lung adenocarcinoma cells A549 is p53/p21-dependent and impairs cellular redox state, mimicking the cisplatin action. A549 cells exhibit decreases in their reducing power and glutathione level after incubation with 5 mM thiamine diphosphate. Thiamine deficiency elevates glutathione in A549 cells. However, the increase is attenuated by 5 mM thiamine diphosphate, p21 knockdown, specific inhibitor of the 2-oxoglutarate dehydrogenase complex (OGDHC) or cisplatin. Authors conclude that thiamine diphosphate and cisplatin affect viability by OGDHC mediated mechanism of the damage of A549 cells, compromising cellular redox state under the control of p53-p21 axes.

This paper describes a novel concept of activating the 2-oxoglutarate dehydrogenase by supplementation with its cofactor thiamine diphosphate to target cancer cell.

Minor suggestions:

Authors should consider revising the Figure 1 and Figure 2 texts. The subtitles such as Control and Thiamine deficiency are too big - reduce the font size. Also reduce the font size for cell types.

Meanwhile, some text in Figure 2 is not legible such as Y-axes labels which must be increased in size.

Figure 3 "r" and "p" values must be presented in black bold legible font such as Arial.

Figure 4 and 5: Reduce the subtitle font size and improve legibility of the small text such as statistical data at the bottom.

Reviewer 3 Report

This paper is very well writen even if being very dense with a quantity of references that fit more a review than a research paper.

Publish as is comes !

NB: the only small improvement will be to reduce a bitteh number of references

Reviewer 4 Report

The authors demonstrated that activation of mitochondrial 2-oxoglutarate dehydrogenase by cocarboxylase is p53-p21 dependent and impairs cellular redox state. The topic is very interesting and the work is very well done and described I have only a few observations:

I think that the demonstration of p53 activation following ThDP supplementation will directly demonstrate the  involvement of this protein in the mechanism, since its name is present in the title.

I think that, the suggestion of thiamine, or its derivatives, supplementation as anticancer therapy  will be better supported showing that the mechanism described is in common with other p53 wild type cancer models.

Round 2

Reviewer 1 Report

See my comments to the editor

Author Response

What is the relevance that activation of OGDH is p53/p21-dependent?

This is considered in the Introduction and Discussion, summarized in the first and last sentences of the abstract as:

“ Genetic up-regulation of mitochondrial 2-oxoglutarate dehydrogenase is known to increase reactive oxygen species, being detrimental for cancer cells…. High ThDP saturation of OGDHC compromises redox state of A549 cells under the control of p53-p21 axes. “

Why is this activation impairing cellular redox state?
As we write in the discussion, this is because overactivation of OGDHC by cocarboxylase impairs the glutathione and redox homeostasis. In particular, because OGDHC is known to produce ROS and control glutamate metabolism

Why does this mimick cisplatin action and why is this relevant
(cisplatin is an anti-cancer drug whereas cocarboxylase is an endogenous
compound.

We employ a wide-spread approach to assess the suggested ThDP action through the action of a drug with the well-studied mechanism. The approach indicates the competitive action of cisplatin and cocarboxylase, which provides evidence for increased ROS generation by the cells with ThDP-overactivated OGDHC. This is relevant because cisplatin is toxic to normal cells while ThDP is not.

All these issues are carefully considered in Discussion and Graphicalo abstract

Besides, the authors use a tremendous qamount of self citations, which
is irritating.

We are sorry to irritate the reviewer, but we consider 18 out of 79 references (23%) quite acceptable, especially taking into account that we refer to many original and pioneering methodical approaches published by our group. Besides, the authors include leading scientists in the research on OGDHC, to which the manuscript subject is dedicated. This judgement is exemplified by a contract of Elsevier with Dr. Bunik to prepare a chapter on OGDHC for Elsevier encyclopedia.
